# Synthesis and Characterization of Boron Thin Films Using Chemical and Physical Vapor Depositions

Bart Schurink [1,*], Wesley T. E. van den Beld [1], Roald M. Tiggelaar [2,*], Robbert W. E. van de Kruijs [1] and Fred Bijkerk [1]

1  Industrial Focus Group XUV Optics, MESA+ Institute, University of Twente, P.O. Box 217, 7500 AE Enschede, The Netherlands; w.t.e.vandenbeld@utwente.nl (W.T.E.v.d.B.); r.w.e.vandekruijs@utwente.nl (R.W.E.v.d.K.); f.bijkerk@utwente.nl (F.B.)
2  NanoLab Cleanroom, MESA+ Institute, University of Twente, P.O. Box 217, 7500 AE Enschede, The Netherlands
*  Correspondence: bart.schurink@micronit.nl (B.S.); r.m.tiggelaar@utwente.nl (R.M.T.)

**Abstract:** Boron as thin film material is of relevance for use in modern micro- and nano-fabrication technology. In this research boron thin films are realized by a number of physical and chemical deposition methods, including magnetron sputtering, electron-beam evaporation, plasma enhanced chemical vapor deposition (CVD), thermal/non-plasma CVD, remote plasma CVD and atmospheric pressure CVD. Various physical, mechanical and chemical characteristics of these boron thin films are investigated, i.e., deposition rate, uniformity, roughness, stress, composition, defectivity and chemical resistance. Boron films realized by plasma enhanced chemical vapor deposition (PECVD) are found to be inert for conventional wet chemical etchants and have the lowest amount of defects, which makes this the best candidate to be integrated into the micro-fabrication processes. By varying the deposition parameters in the PECVD process, the influences of plasma power, pressure and precursor inflow on the deposition rate and intrinsic stress are further explored. Utilization of PECVD boron films as hard mask for wet etching is demonstrated by means of patterning followed by selective structuring of the silicon substrate, which shows that PECVD boron thin films can be successfully applied for micro-fabrication.

**Keywords:** boron; thin film; CVD; PVD; deposition rate; thickness uniformity; intrinsic stress; chemical resistance

## 1. Introduction

Boron (B) is a low-abundance earth element and differs from its group elements in the periodic system, as its high hardness yet low brittleness are unique elemental properties for a metalloid. Boron finds its major use in reinforcement and insulation of industrial fiber plastics, metal alloys and ceramics. Moreover, boron has applications in the semiconductor industry, where it is utilized for implantation and solid state diffusion by thermal processing and allows for the functionalization of materials and realization of alloys to enhance electrical [1–8], chemical [9–17] and optical properties [18–22]. The specific deposition techniques applied in the semiconductor industry are physical vapor deposition (PVD), such as evaporation and sputtering, and chemical vapor deposition (CVD).

Besides the use as dopant, boron in the form of a thin film has the potential to be a suitable material for micro-machining. Physical characteristics as tunable stress and low roughness are especially useful to tailor properties of free-standing boron thin films (membranes) or cantilever structures. For realizing such boron structures typically fabrication processes are involved, which are based on the etch selectivity during wet chemical processing. Current semiconductor and micro-fabrication technologies are primarily based on processes utilizing selective etching and deposition of thin films with a thickness in the range of tens to a few hundred nanometers. These thin films, commonly (poly)silicon,

silicon dioxide and silicon nitride, are typically selectively structured by a selected, rather conventional and well-known, set of etchants, including hydrofluoric acid (HF) based solutions and strong alkaline solutions, such as potassium hydroxide (KOH) and tetramethylammonium hydroxide (TMAH). Thus, introducing a thin film material which is resistant to these commonly used etchants is beneficial. Obviously, such a chemical resistant film can also be used to protect an underlying film/substrate against wet chemical etching. For this particular function the presence of defects, such as pinholes and thickness in homogeneities, may cause defects in the underlying device film/material, or may result in undesired etching, for which reason, defects in boron thin films have to be avoided.

In general, the growth of a thin film depends mainly on the 'energy of formation', which results in a film's specific characteristics. In practice, every deposition type (either CVD or PVD based) covers a specific energy window, which is defined by practical limits of a deposition system and by its operational deposition temperature, pressure and boron source. We selected a number of common, in-house (thus, locally accessible) available chemical and physical vapor deposition techniques with a boron source, i.e., magnetron sputtering (MS), electron-beam evaporation (Evap), plasma enhanced CVD (PECVD), thermal/non-plasma CVD (NPCVD), remote plasma CVD (RPCVD) and atmospheric pressure CVD (APCVD).

The realized boron films are studied concerning a number of characteristics, with the annotation that the parameter space was not explored extensively, but rather concerned conditions which were found to yield adequate results from a number of separate, local optimizations. Results are therefore machine and material dependent. In fact, similar to the deposition techniques employed in this work to realize boron thin films, over the last decades for a variety of thin films/materials and/or deposition techniques investigations/characterizations/optimizations were performed by MESA$^+$ affiliated academic research groups [23–39]. This work is dedicated to research in the realization of boron thin films and focuses on typical film characteristics of films obtained with various deposition techniques.

The boron film characteristics studied include deposition rate, uniformity, roughness, stress, composition, chemical resistance and defectivity. The deposition rate is of interest from a productivity point-of-view. Moreover, the deposition rate in combination with the background pressure can potentially relate to the number of impurities built into the film (such as oxygen content). The thickness uniformity is of interest for wafer-scale sized structures (or any other large surface substrates) in which the application of the film scales with thickness, mainly optical or electrical designs. The roughness is also relevant, since it influences surface interactions like wettability and adhesion. The film intrinsic stress is especially of importance in case the thin film is fabricated into a free-standing structure, as in a pellicle, membrane or cantilever. Furthermore, the intrinsic stress has a direct influence on the mechanical properties of the substrate and already present layers resulting deflection and strain. The composition of the boron film is important in thermal or mechanical applications, where impurities affect the material stability as well as strength. Lastly, the number of defects and chemical resistance are important if the boron film is applied as protective coating for underlying layers in combination with a reactive environment (both during fabrication as well as in applications).

In follow-up to this brief exploration of boron thin films deposited by different methods and the resulting film characteristics, we focus on the PECVD method in more detail based on the found boron film properties. Several PECVD parameters—i.e., the precursor flow, the pressure, the plasma power and the deposition temperature—were varied to investigate their influences on the deposition rate, thickness uniformity and intrinsic stress. Finally, these PECVD thin films are evaluated for their suitability as a thin film material for microfabrication processes, such as for usage as masking or structural layer.

## 2. Materials and Methods

All boron depositions are performed on 100 mm diameter single-side polished p-type, (100)-oriented silicon substrates (Si-Mat, Kaufering, Germany) with a thickness of $525 \pm 25$ μm and a resistivity of 5–10 Ωcm. In this work, the deposited films have been reproduced at least 2 times, and no discrepancies were found. In case of PECVD, NPCVD and APCVD, prior to boron deposition any native oxide was removed from the substrates by immersion in 1% hydrofluoric acid (HF, 1 min; Technic, Balve, Germany), followed by quick dump rinsing in demineralized water and spin drying. This removal step was not performed prior to magnetron sputtering and RPCVD due to logistic challenges.

To ensure a minimal introduction of defects onto the surface of substrates prior to boron deposition, the silicon substrates were handled with care in the ISO 5 class clean-room, to prevent damage and the introduction of particles influencing the boron thin film characterization on defects. In addition, the effect of the removal of native oxide was investigated as this could potentially introduce the defects. A comparison was made between out-of-the-stock-box substrates and substrates of which the native oxide was stripped. This evidenced that no significant number of particles were added to the substrates upon applying the procedure for native oxide removal prior to boron thin film depositions. More specific, five substrates of both types were inspected by dark field (DF; with a tungsten halogen light source; MX-61, Olympus, Leiderdorp, The Netherlands) optical microscopy and bright spots on each surface were counted at a 100× magnification in two perpendicular directions crossing the wafer center. Typical densities of particles on out-of-the-stock-box wafers and wafers of which the native oxide was removed were found to be 0.1 and 0.3 particles/mm$^2$, respectively. In an identical manner the defects after the boron depositions were observed and counted. By etching the underlying silicon even non-observable pinholes after deposition could be visualized.

### 2.1. Deposition Methods and Settings

In Table 1 an overview is given of the boron deposition methods used in this work, including applied settings. More details about the CVD and PVD systems can be found in Appendices A and B.

**Table 1.** Settings of the boron deposition methods as used in this work, i.e., plasma enhanced CVD (PECVD), non-plasma CVD (NPCVD), remote plasma CVD (RPCVD), atmospheric pressure CVD (APCVD), magnetron sputtering (MS) and evaporation (Evap).

| | CVD Methods | | | | PVD Methods | |
|---|---|---|---|---|---|---|
| | **PECVD** | **NPCVD** | **RPVCD** | **APCVD** | **MS** | **Evap** |
| Set point substrate deposition temperature [°C] | 300, 350, 400 | 350 | 450 | 700 | room temp. | room temp. |
| Deposition pressure [mTorr] | 1000, 1400, 1800 | 210 | 750 | $7.5 \times 10^5$ | 0.6 | $2.4 \times 10^{-4}$ |
| Precursor and flow [sccm] | $B_2H_6$ (5% in Ar) 25, 50, 75 | $B_2H_6$ (5% in Ar) 100 | $B_2H_6$ (0.5% in Ar) 200 (+150 sccm Ar carrier) | $B_2H_6$ (0.2% in $H_2$) 490 | >98% B 1–2% C (30 sccm Ar) | 99.5% 3–8 mm crystalline B pieces |
| Source power [W] | 30, 100, 200, 240, 400 (RF) | - | 2000 (remote) | - | 500 (1A (DC) 560–580 V) | - (72 mA 10 kV) |

## 2.2. Annealing

Annealing of the boron films was performed in an atmospheric tube furnace at 650 °C (Tempress, Vaassen, The Netherlands) under a nitrogen ($N_2$) flow for 1 h. This specific temperature was chosen to accomplish out-diffusion of bonded or embedded 'free' hydrogen from the boron films, but also to prevent the formation of volatile boron species as well as to minimize diffusion of boron into the silicon substrate.

## 2.3. Analysis

The deposited film thickness (including its uniformity) was measured by X-ray reflectometry (XRR) (Empyrean laboratory diffractometer; Malvern Panalytical, Almelo, The Netherlands) at 3 locations on the 100 mm substrate, namely, approximately in the center (0 mm) and 20 and 40 mm towards the wafer edge.

The surface roughness of the boron films was measured by atomic force microscopy (AFM) in 'small area tapping' mode (Dimension Edge; Bruker, Billerica, MA, USA) for a surface area scan of $1 \times 1$ μm. Analysis of the roughness as root mean square (RMS) was performed using Gwyddion (v2.44) SPM software.

Mechanical deflection of substrates was measured by white light interferometry (WLI) (NewView 7200; Zygo, Middlefield, CT, USA) before and after deposition and, by applying Stoney's equation [40], the intrinsic stress in the boron films was calculated (details of the protocol used for stress calculations based on WLI can be found in [41]). The center point of each substrate was measured with a $1\times$ magnification (field of view: 120 mm$^2$).

The composition of the deposited boron films, including oxygen content, was studied with X-ray photoelectron spectrometry (Thermo-Fisher Scientific, Bleiswijk, The Netherlands). Ar$^+$ sputter depth profiles experiments were performed (1 keV sputter energy, 2.5 mm $\times$ 2.5 mm sputter spotsize) and resulting data were analyzed using Thermo Scientific Advantage System software (Advantage 5.968).

The elemental composition, including hydrogen, of some films was analyzed by RBS (Rutherford back-scattering spectrometry) and ERD (elastic recoil detection) by Detect99 (Eindhoven, The Netherlands). Measurements were carried out with a 2000 keV He$^+$ beam. ERD was performed with the sample tilted by 75° and the detector at a recoil angle of 25°. RBS was performed under perpendicular incidence and, where possible, in channeling mode, to reduce the background under the light element features. For RBS two detectors were used at a variable angle and a fixed scattering angle of 170°.

The crystalline properties of a RPCVD boron film were investigated using grazing incidence X-ray diffraction (GI-XRD) analysis (Empyrean X-ray diffractometer with a Cu-K$\alpha$ source (1.5406 Å); Malvern Panalytical, Almelo, The Netherlands) under the condition $\omega_{in}$ = 1°. The XRD-spectrum (Appendix C) shows an amorphous film (as for all films).

The chemical resistivity was investigated by wet etch tests: samples with the deposited boron films were immersed in buffered hydrofluoric acid (HF:NH$_4$F 7:1; Technic, Balve, Germany) solution for 10 min and a 25 wt.% tetramethylammoniumhydroxide (TMAH, Technic, Balve, Germany) solution at 90 °C for over 5 h. Prior and post to wet chemical treatments, the thickness of the boron films was measured with XRR to obtain the etch rate, and thereby information about the resistivity of the boron films for the mentioned wet chemicals.

## 3. Results and Discussion

The deposited boron films have been analyzed on their use as material for surface and bulk micro-machining. In the following subsections, obtained results regarding film thickness and non-uniformity, roughness, intrinsic stress, elemental composition, resistance to certain wet chemicals and number of defects are discussed. An overview of the investigated thin film properties of all deposition methods is shown in Table 2. A detailed discussion of these properties follows in Sections 3.1–3.9.

**Table 2.** Characteristics of boron thin films realized in this work by various CVD and PVD methods, i.e., plasma enhanced CVD (PECVD), non-plasma CVD (NPCVD), remote plasma CVD (RPCVD), atmospheric pressure CVD (APCVD), magnetron sputtering (MS) and electron-beam evaporation (Evap). Shown data are obtained from the center of the wafer. The thickness non-uniformity is calculated as percentage from the boron thickness measured at the center of the substrate relative to the thickness at 40 mm from the center. The XPS atomic percentages indicate those at half-thickness of the layer ([1]: 'No' implies no loss in material thickness).

| | | CVD Methods | | | | PVD Methods | |
|---|---|---|---|---|---|---|---|
| | | PECVD | NPCVD | RPVCD | APCVD | MS | Evap |
| Thickness (center) [nm] | | 47.4 | 35.1 | 34.1 | 18.2 | 31.2 | 52.8 |
| Deposition rate [nm/min] | | 14.4 | 0.6 | 9.0 | 0.3 | 2.7 | 8.3 |
| Thickness non-uniformity | | 19% | 58% | 20% | <5% | <2% | 11% |
| Roughness RMS [nm] | | 1.0 | 1.3 | 1.6 | 0.4 | 0.2 | 0.2 |
| Stress [GPa] | | −0.2 | 0.3 | 0.8 | −2.0 | −4.4 | 0.4 |
| Stress post to 650 °C anneal [GPa] | | 0.8 | 0.7 | 1.4 | −1.8 | −0.4 | 1.5 |
| Elemental comp. [at%] | B | >99 | 95 | >99 | 92 | 98 | 97 |
| | O | <1 | >3 | <1 | <1 | 0.5 | >3 |
| | C | <1 | <2 | <1 | <1 | 1.5 | <1 |
| | traces | - | - | - | Si (>6%) | Ar, metals | Si, metals |
| Etch rate [nm/min] | TMAH | No [1] | 0.1 | No [1] | No [1] | No [1] | No [1] |
| | BHF | No [1] | 25 | No [1] | No [1] | No [1] | No [1] |
| Number of defects [#/mm$^2$] | | 0–5 | 5–10 | 5–10 | 15–20 | 15–50 | 50–100 |

### 3.1. Thin Film Thickness

The deposition rates of the different boron films cover different ranges, resulting from local process development (see Table 2). In fact, PECVD has the highest deposition rate, which is almost 50 times the deposition rate of APCVD. Interestingly, the boron deposition rate of APCVD shows a small decrease within the first 15 nm of the film (i.e., from 0.4 nm/min to 0.3 nm/min), followed by a sharp decay for thicker layers (i.e., down to 0 nm/min), which limits the practical deposition thickness of this method to about 18.5 nm. The reason for this self-limiting thickness effect is related to the composition of this specific film. Namely, XPS analysis showed six atomic percent of silicon throughout the deposited film (see Figure 1 in Section 3.5), which indicates that boron deposited on the surface immediately intermixes with the silicon substrate. As the film thickness increases, more silicon will diffuse through the deposited boron film, resulting in a constant/fixed silicon content throughout the boron film. The diluted boron precursor could play a role in this ongoing intermixing of boron and silicon and the restriction on the maximum achievable thickness. Initially, the film growth is reaction rate limited, followed by a diffusion-limited growth rate when the film is around 15 nm. Hereafter, at 18.5 nm, the growth rate is believed to reach a saturation regime. It is likely that both diffusion through the growing boron layer and processes at the boron–silicon interface have a mixed effect in both a linear and parabolic regime [42].

### 3.2. Thickness Non-Uniformity

The achieved thickness non-uniformity is mostly related to the deposition chamber geometry and design. For PVD the substrate rotation and relatively large target to substrate

distance benefit the uniformity of the deposited boron films. In contrast to PVD, for CVD deposition systems several geometrical system aspects are critical for the thickness non-uniformity, such as the substrate heater, the size of the precursor showerhead, the plasma electrode size and target-substrate distance. The NPCVD system displays the largest thickness non-uniformity (i.e., 58%), which is due to exposure of the substrate to non-preheated gas. This negatively influences the temperature homogeneity at the substrate surface, which, in turn, affects the local deposition rate. A better uniformity is found for the PECVD system: here influence of the plasma allows for the formation of reactive species towards the surface, resulting in a better thickness uniformity. The RPCVD system shows a thickness non-uniformity comparable to the PECVD system. The APCVD system produces boron thin films with a thickness non-uniformity of less than 5%, which is because the deposition chamber is optimized for epitaxial growth processes. In general, our PVD boron films exhibit a better thickness uniformity than CVD-based boron films.

### 3.3. Roughness

Film roughness can be of importance for several applications, and it is found that there is a clear distinction between the PVD and low temperature (LT; <500 °C) CVD depositions (see Table 2; representative AFM-images are shown in Appendix D). The surface roughness of these LT-CVD films is approximately $7 \pm 1$ times larger than the roughness of the PVD films. This difference is mainly caused by the formation of boron clusters of which the initial size scales with temperature [43]. This is supported by the roughness measured on the APCVD film, which is only two times that of the PVD but has two times the deposition temperature of the PECVD [30]. Furthermore, the deposition rate has an impact on the initial cluster size: the higher the flux of atoms/molecules arriving on the surface per unit of time, the larger the clusters grow, and hence the higher the roughness of the deposited film [44]. Moreover, the LT-CVD thermal energy is sufficient to activate the terminated hydrogen of the silicon surface, but restricts mobility, preventing the formation of a continuous/closed film in the initial growth phase, promoting local growth which could increase the film roughness [30].

### 3.4. Intrinsic Stress

In general, stress development during thin film formation correlates to the deposition rate, temperature and film microstructure, giving rise to complex interactions that cannot be fully explained based on solely the intrinsic stress of the boron thin film. Nevertheless, based on the nature of the deposition, the type of this stress, i.e., compressive or tensile, can be explained. In case of CVD films, the first atomic layers of boron will intermix with the silicon substrate if the native oxide is removed (in the case of PECVD, NPCVD and APCVD) and the thickness of this intermixed layer depends on the deposition energy and is increased by temperature and plasma. This intermixing induces thin film relaxation, which adds compressive stress to the growing film. In contrast, the RPCVD has a higher stress compared to the other LT-CVD films, since here, the native oxide layer was not removed, and thus intermixing and relaxation is not expected to take place. In terms of temperature, at <600 °C the (partial) decomposition of $B_2H_6$ results in the formation of boron films without significant intermixing, while at >700 °C the deposited boron has sufficient energy to form a silicide [45]. This is evidenced by the 1.8 GPa compressive stress of the APCVD film, which is assumed to be the 6 at% of silicon in the film (see Table 2). An even higher compressive stress is found for the sputtered film ($-4.4$ GPa). The incoming boron atoms have sufficient energy (or rather momentum) for continuous intermixing with the first atomic layers on the surface [46], either the recipient film (native oxide on silicon) or the already deposited film [47]. In the literature, this phenomenon is referred to as ion-peening, knock-on implantation, recoil implantation or sub-plantation, and is responsible for densification of the films and can result in phase transformations [48,49] and high compressive stresses [50]. In comparison with CVD, the entire substrate and growing film are thermally energetic and allow for continuous intermixing (diffusion) over all atomic

layers. In sputtered boron films this surface intermixing, implantation and reordering of the relative small boron atoms result in a higher density film than a CVD film. In electron-beam evaporation, a significant lower energy is expected compared to magnetron sputtering, which leads to less atom intermixing, implantation, reordering, and thus, a lower density is expected [51]. Hence, the found stress of 0.35 GPa in evaporated boron thin films is in agreement with the literature: once the initially formed islands coalesce, relaxation occurs and further material added to the film yields a tensile stress [52].

Analysis of the deposited boron films is performed post to the $N_2$ annealing. Annealing does not reduce the thickness but induces the formation of surface oxide ($BO_x$). The latter cannot explain the large stress changes as reported in Table 2. In fact, after annealing, the boron films become less compressive or (more) tensile, except for the APCVD film, which is not affected by annealing at temperatures below its deposition temperature of 700 °C. The origin of the observed stress change after annealing in this study is not found, but mechanisms at play could include diffusion and intermixing of the boron with the silicon, thin film relaxation, densification or out diffusion of hydrogen.

For the sputtered boron films, a stress shift of 4 GPa is observed after annealing, as can be seen in Table 2. This large shift in stress cannot be explained by the XPS data, since there are no significant compositional changes. In addition, we found that the deposition of different thicknesses of sputtered boron film results in a reproducible stress. Combining these two findings would suggest that no material is removed from the film upon annealing and interface effects do not play a significant role, as of which both cannot explain the 4 GPa stress change. The XRR data pre and post annealing (not shown) shows a slight contrast increase, suggesting that no significant diffusion of boron through the native oxide layer into the silicon substrate has occurred. Comparable experiments from the literature [53–55] support this finding of a native oxide functioning as diffusion barrier from chemical deposited boron sources. Therefore, we believe that the stress change is the result of film relaxation and densification [56,57].

The origin of the stress change (0.4 to 1.5 GPa) in evaporated boron films after annealing is most likely caused by densification. No significant potential volatile impurities (including hydrogen and oxygen species) are incorporated in this film (XPS data, RBS and ERD data), which might be due to the relative high pressure ($7.6 \times 10^{-6}$ mTorr) during deposition. Furthermore, the intrinsic stress is already tensile, indicating the low energetic nature of thermal evaporation, and hence the lack of the typical effects in sputtering (regarding ion-peening and implantation) to densify the evaporated thin film during growth [51,58,59].

For the PECVD, NPCVD and RPCVD films, we assume that the deposition temperature [45] and/or plasma energy [60] is not sufficient to fully dissociate the $B_2H_6$ precursor based on the literature and our analysis. Consequently, during deposition the $B_2H_6$ precursor dissociates into borane ($BH_3$), which adheres to the surface, resulting in one or two hydrogen atoms per boron atom being incorporated into the film. During annealing at temperatures above the deposition temperature, this built-in hydrogen will diffuse out, tentatively resulting in a more tensile stress [61], in agreement to the results presented in this paper. RBS and ERD measurements on the sputtered and PECVD boron films support this hypothesis, since the hydrogen content in as-sputtered film is below 2%, whereas non-annealed PECVD films contain up to 16% hydrogen.

### 3.5. Elemental Composition

In Figure 1, XPS depth profiles of boron films realized by three different deposition methods are shown. XPS depth profiling showed that the sputtered boron film contains around 98 at% boron (B), 1.5 at% carbon (C) and 0.5 at% of oxygen ($O_2$) and argon (Ar) and metal traces (as generally expected for a sputter-deposited film). A comparable composition is found for the evaporated film, i.e., 97 at% B, <1% C and >3% $O_2$, as well as silicon (Si) and metal traces. The PECVD film contains over 99 at% of B and only a slight number of residual elements and the RPCVD films have a similar boron content. The NPCVD film

has a significant lower boron concentration (95 at%) as well as a high oxygen content (over 3 at%) and less than 2 at% of C as well as trace elements. The relative high oxygen content in the NPCVD film can be attributed to the relative high background pressure and low deposition rate this oxygen will be embedded into the film. In fact, an even a lower B percentage is found in the APCVD film (less than 92 at%); however the APCVD film has no increased oxygen level. Due to the relatively high APCVD temperature, B and Si intermix during the film formation, as explained in Section 3.1.

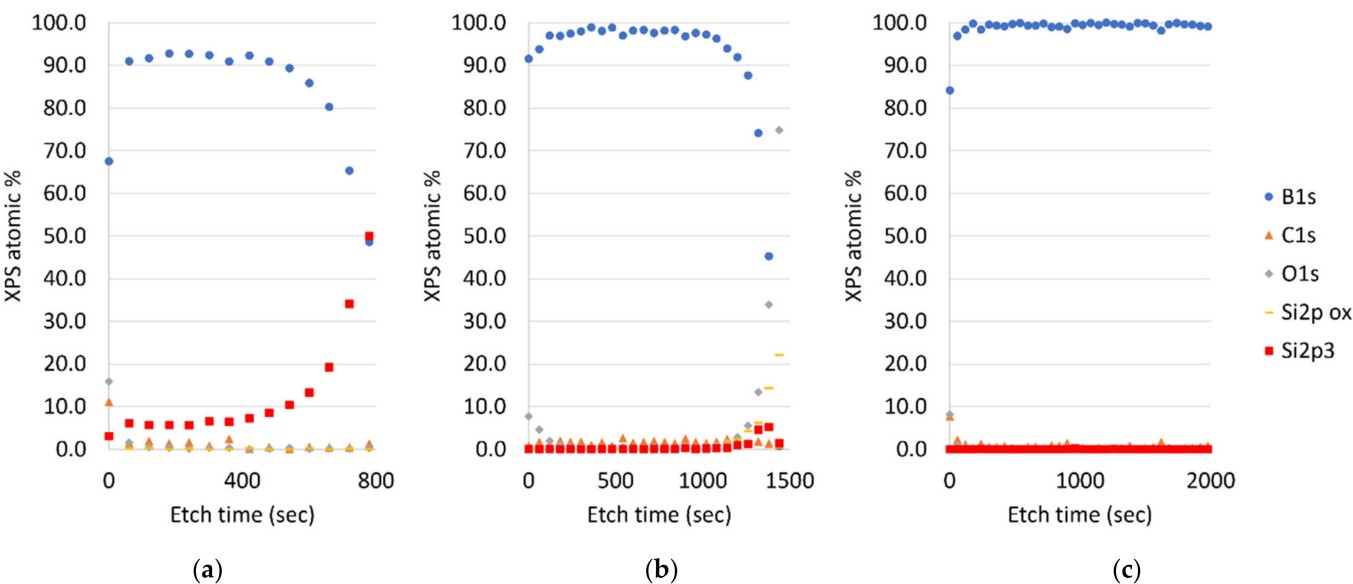

**Figure 1.** XPS depth profiles of boron thin films realized in this work by (**a**) atmospheric pressure CVD, (**b**) magnetron sputtering, (**c**) plasma enhanced CVD.

### 3.6. Chemical Resistance

We found that a boron thin film, when free of contaminants, is chemically resistant to the wet etchants TMAH and BHF, which are primarily used to etch, respectively, Si and oxide films. From chemical resistance tests, such as immersion of substrates with boron thin films in TMAH or BHF solutions, it follows that the native oxide (1–2 nm thick) on the deposited boron films is removed rapidly from the films upon immersion in these etchants. As can be seen in Table 2, none of the investigated boron films is etching in TMAH or BHF, except for the NPCVD boron film. We believe that an increased oxygen concentration to approximately 3 at%, as in the NPCVD boron layer, leads to etching of such film in TMAH (ca. 0.1 nm/min) and BHF (ca. 25 nm/min). In contrast, the APCVD film, with over 6 at% of Si, does not etch upon exposure to TMAH for long etch times (>5 h). This is in agreement with boron doped silicon etch stops; although such etch stops have significant lower percentages of boron compared to the APCVD film [13,62].

A chemical resistance of the boron film is needed in case it is utilized as a hard mask (or structural material/layer) for selective wet etching of silicon, silicon nitride or silicon dioxide. Based on our findings only the NPCVD film seems to be unsuitable for this application due to its low chemical resistance.

### 3.7. Amount of Defects in the Films

Besides that a hard mask or structural layer composed of boron should not etch in specific etch solutions (in order to protect the underlying material), the boron film also has to be continuous, i.e., without any defects, such as pinholes or clusters containing (or composed of) dissolvable or etchable material. To qualify this aspect of the various boron films, DF microscopy is used for the detection of defects in size well below the diffraction limit of the maximum magnification (DF microscope with 100× objective). In

practice, the size-detection limit of the defects was found to be around 100 nm in diameter, which was confirmed by scanning electron microscopy. It is noted, however, that defect levels are known to be largely determined by the general cleanliness conditions of the local process environment.

Although every film contained some defects, not every defect turned into a pinhole, allowing for access to the material below the film. In more detail, a particle transferred onto the surface can be encapsulated and become a part of the final film composite. This encapsulation is considered to be different for PVD compared to CVD. This is proven by our chemical resistivity experiment, where the boron film is exposed to TMAH for 5 h. The silicon underneath a pinhole is etched, which makes detection of these defects convenient by their size (since these defects become larger in size than the pinhole width) and their characteristic inverted pyramidal shape. The difference in encapsulation is considered to be the effect of the PVD line-of-sight site phenomenon, in which particles will not be fully enclosed and remain exposed to the environment after the deposition. This in contrast to CVD, in which every surface is potentially coated, including present particles, resulting in a full encapsulation, even if the size of the particle ranges beyond the thickness of the film.

The highest number of defects is found in the evaporated boron films, locally up to 100 per mm$^2$ (see Table 2), where sputtered films contain ca. 15–50 defects per mm$^2$. The relative high number of defects in these films was expected, since in general PVD systems contain moveable mechanical parts (such as a transport arm, rotating sample stage and shutter(s)). These delaminated "flakes" can shatter into numerous small particles, which tend to be attracted by electrostatic forces towards the substrate.

For CVD-based boron films, the majority of the defects are not expected to be of mechanical origin. Yet particles could originate from the deposition chamber wall from build-up of earlier coatings, especially if different materials with a relatively high intrinsic stress are deposited in a single chamber. Alternatively, film defects can be the result of chemical interactions, viz. boron reacting with residual oxygen in the chamber before absorbing on the silicon substrate. This is supported by the correlation between the amount of defects with the deposition rate: the lower the deposition rate, the larger the amount of defects in CVD-based boron films (Table 2). More specific, the relative high background pressure and low deposition rate results in an increased probability of oxygen species to be incorporated into the boron film.

In summary, a first, local inventory of boron thin films realized by a number of chemical and physical deposition techniques has been made for its use in and suitability for micro-fabrication technology. We characterized their chemical resistance to conventional etchants for silicon, silicon nitride and silicon oxide, as well as absence of pinholes/defects. Except for APCVD boron films, all investigated PVD and CVD boron films possessed this etch inertness, but only the PECVD film had a very low amount of defects. This motivates a further study of the PECVD system (i.e., the precursor flow, pressure, plasma power and temperature) on the boron film, with the aim to improve several film properties, i.e., deposition rate, film thickness uniformity and stress.

### 3.8. Influence of PECVD Parameters on Boron Film Properties

For thin PECVD boron layers a series of more detailed depositions was performed by varying flow, plasma power, deposition temperature and pressure. The used settings for precursor flow, pressure, plasma power and temperature were chosen within the range of the local PECVD system and are listed in Table 1. The deposited films are studied for intrinsic stress and growth properties.

As can be seen in Figure 2, the intrinsic stress of PECVD boron films is affected primarily by the deposition temperature and, to a lesser extent, the RF power. For a deposition temperature of 300 °C a compressive film is obtained, whereas at a deposition temperature of 350 °C, the film has a low or nearly-zero stress. More specific, at 350 °C and at high plasma power (i.e., 120 to 240 W) the films are slightly compressive, whereas a low power (i.e., 60 W and below) in combination with a chamber pressure of 1800 mTorr

yields slightly tensile films. For deposition at 400 °C the films exhibit a tensile stress with values in the range 0.18 to 0.42 GPa. Whereas the intrinsic stress increases with deposition temperature, the stress reduces for increasing plasma power. As can be seen, the PECVD boron films deposited at 400 °C are tensile for the investigated RF power range, the films deposited at 350 °C experience a stress transition (from tensile to compressive) and the films deposited at 300 °C are compressive. This difference in intrinsic stress can be explored to mediate multilayer film stress, as well as to tune the behavior of cantilever structures (or free-standing films) in microsensors and microactuators for optical and electrical applications.

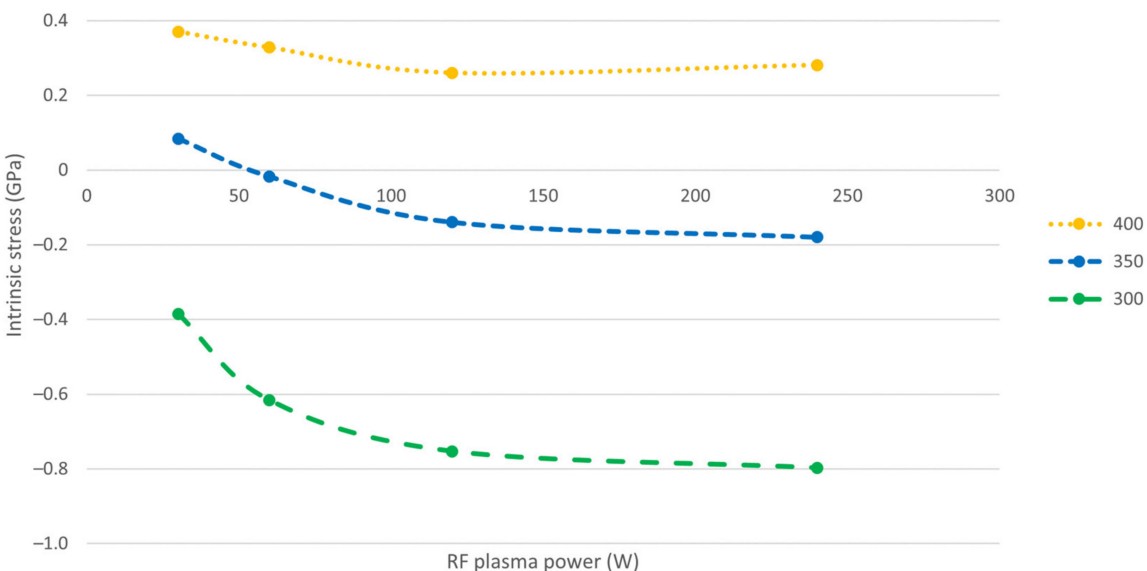

**Figure 2.** Intrinsic stress in PECVD boron thin films deposited at different temperatures and RF plasma powers.

In order to better understand the influence of the main deposition parameters (plasma power, pressure and flow rate) in the boron PECVD process, the Arrhenius activation energies are calculated by using the temperature dependent deposition rate in the range of 300–400 °C. For the diborane precursor in an inert carrier gas ($N_2$ or Ar), the boron deposition is mainly controlled by the heterogeneous reaction of the decomposed precursor ($B_2H_6(g) \rightarrow BH_3(g)$) with free Si or B surface sites ($BH_3(g)$ + Si(s) or B(s)) [63,64]. Therefore, the found activation energies are associated with this reaction.

In general, for PECVD the plasma is known to reduce the activation energy compared to thermal CVD [65]. This takes place by adding energy to vibrational excitations of the diborane precursor molecules [66]. The higher the reactor plasma power, the higher the plasma ion density, and effectively more energy is supplied for activation of the precursor [67]. As a consequence, for higher reactor plasma powers a lower effective activation energy is expected. However, this effect of reduction in activation energy will saturate at higher plasma powers, since the ion density then saturates. This effect is confirmed by our data shown Figure 3: at first the activation energy decreases rapidly because of molecule activation via the plasma, followed by saturation for increasing plasma powers because of the saturating plasma ion density.

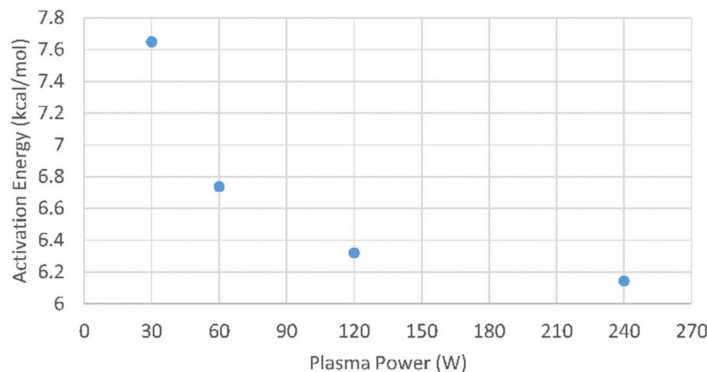

**Figure 3.** The effective activation energy for PECVD boron films as function of plasma power.

Generally, in plasma systems an increasing pressure leads to a lower plasma ion density, which causes effectively less energy to be supplied to the precursor molecules [67,68]. Based on this, the effective activation energy is expected to increase for increasing pressures. This trend is confirmed by our data shown in Figure 4.

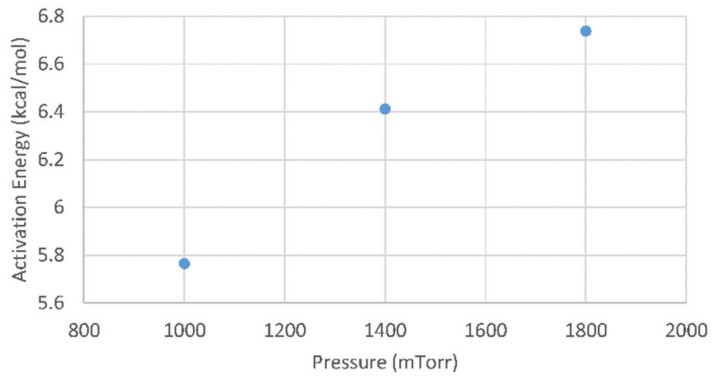

**Figure 4.** The effective activation energy for PECVD boron films as function of reactor pressure.

Lastly, the effective activation energy is determined as a function of the flow rate: a proportional decrease of activation energy for increasing flow rates is found (data not shown). This can be explained by the increased gas convection that occurs for increased flow rates, which causes more activated precursor species to arrive at the substrate [69].

The determined effective activation energies can be compared with the literature values. Mohammadi et al. report an activation energy of 2.1 kcal/mol for boron deposition using an inert carrier gas (i.e., $N_2$ for this reported value and Ar for our experiments) [30], which is significantly lower than the values which we determined. The difference can be explained by the fact that the diborane partial pressure as used in our work (50–90 mTorr) is one order of magnitude higher compared to the reported values, and higher pressures cause an increased effective activation energy (see also Figure 4).

### 3.9. Use of PECVD Boron Films As Mask Layer/Structural Material in Micro-Machining

In this section we demonstrate the potential of PECVD boron thin films for surface and bulk micro-machining, as also recently reported by Liu [70]. A PECVD boron thin film was deposited on a 100 mm diameter silicon wafer (as described in Section 2) at 350 °C, 1800 mTorr, 50 sccm $B_2H_6$ (5% in Ar) and 60 W RF. On this film positive photoresist (Olin 907/17) was spin coated, baked and exposed by UV light through a photomask containing lines of different widths (1–10 μm). Afterwards, the photoresist was developed (OPD 4262) and transferred into a plasma etch system (Alcatel-Adixen AMS100DE DRIE Etcher, Alcatel Vacuum Technology, Annecy, France). The boron film was dry etched for 135 sec (so an effective etch rate of approximately 25 nm/min) using 100 sccm $SF_6$, 20 sccm Ar, 250 W

ICP, 25 W CCP, $1 \times 10^{-2}$ mbar. Ellipsometry was used to confirm the complete removal of the boron at the exposed locations. Following this step, the sample was immersed in a BHF bath for 1 min to remove the native $SiO_2$, followed by stripping of the photoresist with acetone. This was followed by immersion in TMAH (25 wt.% at 90 °C) for 3 min, resulting in anisotropic etching of the exposed silicon using the patterned boron film as a hard mask layer. The result was trenches confined by <111> crystalline silicon planes, which extend underneath the boron film (Figure 5). Careful inspection of Figure 5b reveals that directly underneath the boron thin film a different etch crystal plane is visible, i.e., different from the expected <111> crystalline silicon plane. This plane is related to the dry etch step (used for patterning the boron film), during which also slight etching of the silicon substrate occurred, yielding a trench with vertical side walls. Anisotropic wet etching starts from this trench and its vertical walls give rise to a wet-etched groove with the shown non-<111> planes near the opening in the boron film hard mask [71].

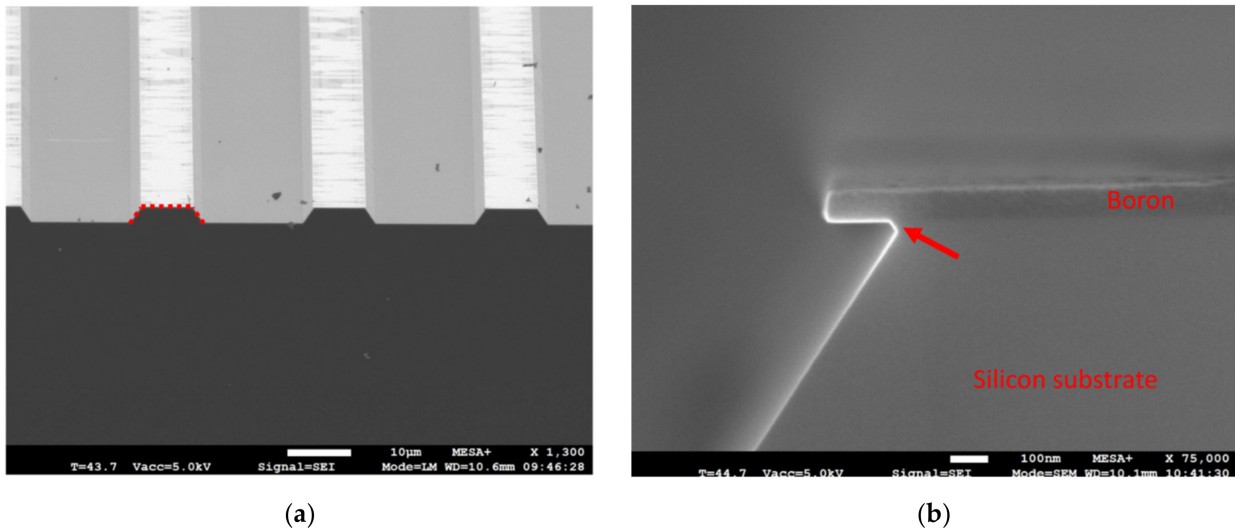

(**a**)  (**b**)

**Figure 5.** SEM images of a 52 nm thick boron PECVD film (350 °C, 60 W RF, 50 sccm $B_2H_6$ precursor, 1800 mTorr) used as hard mask for selective wet chemical etching of silicon: (**a**) SEM image showing 8 μm wide boron lines with a 26 μm pitch; the exposed silicon (Si) was etched by TMAH, resulting in trenches bound by <111> crystalline Si-planes (a red line is added as guide to the eye); (**b**) silicon was also etched underneath the boron film hard mask; the small angular deviation of the silicon <111> crystal planes directly underneath the boron film (indicated with a red arrow) is the result of unwanted plasma etching of silicon during structuring of the boron film with reactive ion etching.

The results described in this work demonstrate that PECVD boron films can be patterned with UV-lithography, dry plasma etching, wet etching (BHF) and the structured film can subsequently be used as a hard mask in an alkaline etch solution by means of which the underlying silicon substrate is selectively structured. These results hold promise for novel surface and bulk micromachining applications, and—upon proper embedding in fabrication processes—PECVD boron thin films may find their use as structural material, hard mask, sacrificial layer, functional film mask and/or boron source in (the realization of) applications such as cantilevers [72], hinges [73], pellicles [41], membranes [74], 3D micro electrode arrays [75], MEMS-based structures [76,77], crystalline silicon nanowires [78,79] or p/n junctions [80,81].

## 4. Conclusions

An overview is given of various properties of thin boron films using different deposition methods. The investigated methods are magnetron sputtering, electron-beam evaporation, plasma enhanced chemical vapor deposition (CVD), non-plasma CVD, remote plasma CVD and atmospheric pressure CVD. The boron films realized show a wide variety

in their deposition rate and uniformity, film roughness, intrinsic stress, elemental composition, number of defects and resistance to conventional wet etchants commonly applied for micro-fabrication. Boron thin films realized by PECVD showed the lowest number of defects. The intrinsic stress of the PECVD film is moderate when compared to RPCVD and magnetron sputtered films. Moreover, its intrinsic stress is tunable from compressive to tensile by means of the deposition temperature (as well as post deposition annealing in $N_2$). Besides the deposition temperature, the pressure, plasma power and reactant flow are found to influence the film formation, deposition rate, thickness uniformity and intrinsic stress. Optimized PECVD settings are applied to demonstrate the properties of a boron film in an exemplary application as an etch masking layer in microfabrication: a PECVD boron film is selectively dry-etched after lithography, followed by anisotropic etching of the underlying silicon substrate by TMAH.

**Author Contributions:** Conceptualization, B.S. and W.T.E.v.d.B.; methodology, B.S. and W.T.E.v.d.B.; validation, B.S. and W.T.E.v.d.B.; formal analysis, B.S., W.T.E.v.d.B. and R.M.T.; investigation, B.S. and W.T.E.v.d.B.; data curation, B.S., W.T.E.v.d.B. and R.M.T.; writing—original draft preparation, B.S., W.T.E.v.d.B. and R.M.T.; writing—review and editing, B.S., W.T.E.v.d.B., R.M.T., R.W.E.v.d.K. and F.B.; supervision, R.W.E.v.d.K. and F.B.; funding acquisition, R.W.E.v.d.K. and F.B. All authors have read and agreed to the published version of the manuscript.

**Funding:** This research was funded by the Netherlands Organization for Scientific Research (NWO), in the frame of the Top Sector High Tech Systems and Materials program, grant number 15357.

**Institutional Review Board Statement:** Not applicable.

**Informed Consent Statement:** Not applicable.

**Data Availability Statement:** The data presented in this study are available in this manuscript.

**Acknowledgments:** We thank our colleagues of the XUV Optics group and MESA$^+$ NanoLab cleanroom for their support in realizing the boron thin films and access to measurement equipment.

**Conflicts of Interest:** The authors declare no conflict of interest.

## Appendix A

In this appendix, details of the CVD systems used for the deposition of boron thin films are given.

PECVD of boron was performed in an Oxford 133 Plasmalab system (Oxford Instruments, Yatton, UK) at a temperature of 350 °C using a 50 sccm diluted diborane ($B_2H_6$; 5% in Ar) flow with a top electrode radio frequency (RF) power of 60 W and a deposition pressure of 1800 mTorr. A second set of PECVD depositions was performed using $B_2H_6$ flows of 25, 50 and 75 sccm at pressures of 1000, 1400 or 1800 mTorr, respectively, and deposition temperatures of 300, 350 and 400 °C. Plasma powers for this set were in the range of 0–240 W and 100–400 W in combination with low frequency (LF) or RF, respectively. Combined LF and RF depositions were performed in sets of either 30 or 240 W LF with 100, 200 or 400 W RF. Data from the first set of PECVD films were used to achieve film thicknesses of approximately 50 nm thickness. This particular thickness was chosen to ensure a closed and continuous film, which is compatible with the used analysis methods and it represents a realistic thickness used in practice (for processing or application).

The Oxford 133 Plasmalab system was also used for the NPCVD depositions of boron, using 100 sccm diluted $B_2H_6$ (5% in Ar) at 210 mTorr and 350 °C (it is noted that the RF power source is not used during NPCVD).

Remote plasma atomic layer deposition (ALD), a way of RPCVD, of boron was performed in a Picosun R-200 remote-plasma reactor (Picosun, Espoo, Finland) from $B_2H_6$ precursor (200 sccm, 0.5% in Ar) in a 150 sccm Ar carrier flow. A plasma power of 2 kW, a process temperature of 450 °C, and reactor pressure of 1 mbar were maintained.

For APCVD boron deposition, an Epsilon 2000 deposition system (ASM International, Almere, The Netherlands) was used. Deposition on a rotating substrate (490 rpm) was

performed at 700 °C at 1 atmosphere pressure using $B_2H_6$ as precursor at a concentration of 0.2% (in $H_2$). Prior to boron APCVD deposition, any remaining native $SiO_2$ was removed from the substrate with an in situ thermal cleaning step of 30 min in a $H_2$ ambient at 900 °C.

## Appendix B

In this appendix, details of the PVD systems used for the deposition of boron thin films are given.

For sputtering of boron films, an Advanced Development Coater (ADC) magnetron sputter deposition system (Leybold, Cologne, Germany) was used. DC-magnetron sputtering was done at room temperature from a 4-inch-diameter tiled B target (>98% purity, Kurt J. Lesker Company, Jefferson Hills, PA, USA) in argon (Ar) atmosphere (30 sccm) at a pressure of 0.6 mTorr (chamber base pressure of $7.5 \times 10^{-6}$ mTorr). The rotation speed was set to 10 rpm, the target-substrate distance was 35 mm and the DC power used was 500 W.

Electron-beam evaporation of boron films was performed on a Balzers BAK 600 (Oerlikon-Balzers, Pfäffikon, Switzerland) system. A rotating carrousel at a distance 50 cm away from the crucible contained eight substrates. The crucible was filled with 3–8 mm diameter pieces of crystalline boron target material. The chamber base pressure was $7.6 \times 10^{-6}$ mTorr, the deposition pressure $2.4 \times 10^{-4}$ mTorr, the applied high voltage 10 kV and the measured current 72 mA, respectively.

## Appendix C

The figure (Figure A1) in this appendix shows an XRD-spectrum of a boron film realized by remote plasma CVD: the film is clearly amorphous.

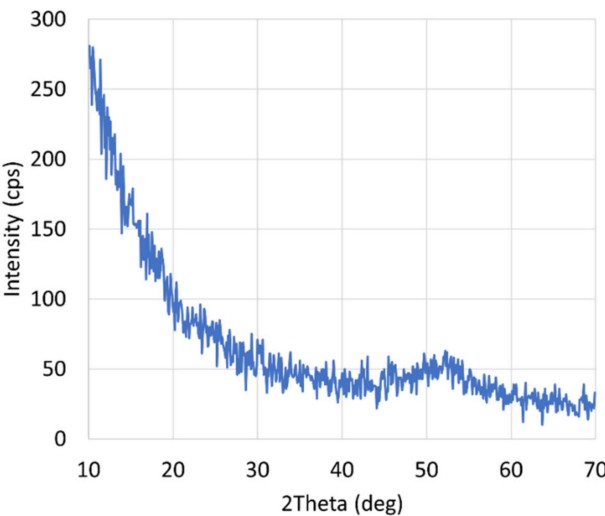

**Figure A1.** Typical XRD-spectrum of a boron thin film realized by RPCVD.

## Appendix D

The figure (Figure A2) in this appendix shows representative images of AFM-scan on boron films realized magnetron sputtering, non-plasma CVD and remote plasma CVD.

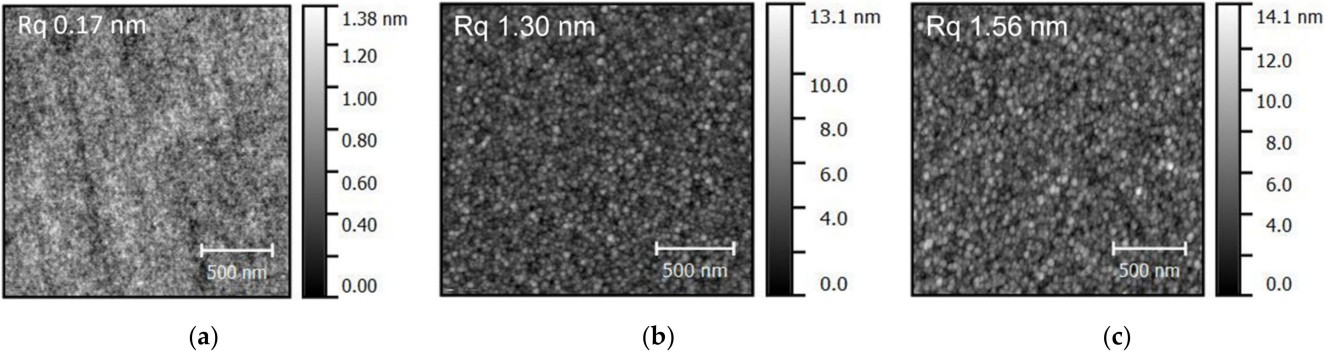

**Figure A2.** Characteristic AFM images (including RMS ($R_q$) roughness numbers) of boron thin films realized by (**a**) magnetron sputtering, (**b**) non-plasma CVD, (**c**) remote plasma CVD.

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
