# Peer review of "Synthesis and Characterization of Boron Thin Films Using Chemical and Physical Vapor Depositions"

_coatings, doi:10.3390/coatings12050685_

Round 1

Reviewer 1 Report

The subject of the manuscript “Synthesis and Characterization of Boron Thin Films using Chemical and Physical Vapor Depositions” by B. Schurink et al. is focused on the characterization of boron thin films deposited by different chemical and physical vapor depositions techniques. This overview ends with the presentation of a focused application of boron films for micro-fabrication processes.   

The manuscript can be accepted for publication after the authors will properly address all the raised queries (in the order they appear in the manuscript):

  1. The authors should try to up-date the references used for the “Introduction” section (only 4 out of 29 date from 2017-2021).
  2. This Reviewer had some difficulties in understanding if the info and the results presented in Tables 1 and 2 actually belong to the authors or from the literature. In any case, proper references should be introduced in a new column attached to the aforementioned tables.
  3. The authors should introduce some info related to the reproducibility of the samples prepared by various techniques.
  4. The morphology of the various synthesized layers is another important parameter which should be also introduced by the authors in the current study. In this respect, comparative SEM images should be included.
  5. Can the authors comment on what is happening with the intrinsic stress at higher deposition temperatures (<500°C?) (Figure 1, page 9).
  6. The authors should carefully check the legend of Figure 4, as the presented images are not AFM ones (but SEM).
  7. The “Conclusions” section should be rewritten and it should include the main findings of the manuscript.

Author Response

The attachment contains our responses to reviewer #1.

Reviewer 2 Report

In this study, the author reported various properties of thin boron films using different deposition methods. The principal idea exhibited in this manuscript looks fine. However, the innovation and depth of understanding of this manuscript is not enough. The manuscript was also not written well. Thereby, I basically agree that this manuscript is not proper for Coatings with the current format.

The comments and issues that require careful consideration:

  1. Some grammar mistakes; such as in line 57-58, lines 72-74. Moreover, the authors should select one of e.g., or i.e., for whole manuscript. After them, we need a comma. Furthermore, what does “ w.r.t." mean in the conclusion part?
  2. Figure 1 format should be changed. It is not good for scientific purpose.
  3. The innovative point of the manuscript is not clear.
  4. Appendix parts should not be included in the manuscript. They are placed in the supporting information.
  5. The references are not up to date.

Author Response

The attachment contains our responses to reviewer #2.

Reviewer 3 Report

The submitted manuscript reports on the preparation of Synthesis and characterization of boron thin films using chemical and
physical vapor depositions.  The aim was to evaluate the physical and mechanical performances of the proposed material. The work performed by the authors is interesting for people in this specific research field. The manuscript itself is well written but has less evidence to support the author's hypothesis and requires major corrections. I have some comments too. The comments are included in the manuscript.

Author Response

The attachment contains our responses to reviewer #3.

Round 2

Reviewer 3 Report

Your effort in the revision is much appreciated.  However, still some minor typo mistakes are available. I have highlighted those. So, it would be nice if you can correct those before the production.

Best wishes for your further work and congratulation on this work.
